# Effects of Gamma Irradiation on Optical Properties of Poly(ethylene oxide) Thin Films Doped with Potassium Iodide

**Ahmed Ali Husein Qwasmeh** [1,*] , **Batool A. Abu Saleh** [2], **Mohammed Al-Tweissi** [1], **Mou'ad A. Tarawneh** [1], **Ziad M. Elimat** [3], **Ruba I. Alzubi** [4] and **Hassan K. Juwhari** [5]

1   Department of Physics, Faculty of Science, Al-Hussein Bin Talal University, Ma'an 71111, Jordan
2   Department of Applied Science, Faculty of Ma'an College, Al-Balqa' Applied University, Al-Salt 19117, Jordan
3   Department of scientific basic Sciences, Faculty of Engineering Technology, Al-Balqa' Applied University, Al-Salt 19117, Jordan
4   Jordan Atomic Energy Commission, Amman 11934, Jordan
5   Physics Department, University of Jordan, Amman 11942, Jordan
\*   Correspondence: qwasmeh@ahu.edu.jo or qwasmeh@hotmail.com

**Abstract:** In this study, the effect of gamma irradiation on the optical properties of thin films of polymer electrolytes was investigated. The thin films were composed of poly(ethylene oxide) (PEO) doped with different concentrations of potassium iodide (KI) salt. The optical absorption spectrum of each film was measured using a UV–Vis spectrophotometer over a range of 300 to 800 nm. The PEO thin-film samples were subjected to gamma irradiation at two different doses of 100 and 200 Gy generated by a Co-60 source. It was found that the optical properties of the thin films were significantly influenced by the KI dopant concentration and gamma irradiation. Increasing both the KI concentration and the gamma irradiation dose resulted in a reduction in the energy gap and an increase in the absorption coefficient, extinction coefficient, refractive index, and dielectric constant of the PEO electrolyte. These findings have potential applications in the optimization of polymers for use in optical devices and energy storage systems.

**Keywords:** gamma irradiation; poly(ethylene oxide); potassium iodide; optical properties; thin films

## 1. Introduction

Polymer electrolytes (PEs) have become a promising group of materials in the field of energy storage and conversion applications. These materials are made by dissolving salts in polymer matrices, which results in the formation of an ionic network within the polymer structure. This unique structure of PEs offers many advantages, such as good thermal and mechanical stability, high ionic conductivity, and wide electrochemical stability windows. Due to these properties, PEs have been extensively studied in recent years as potential candidates for use in various electrochemical devices, including supercapacitors, batteries, and fuel cells [1–5]. Researchers have been investigating ways to further enhance the properties of PEs to meet the requirements of these applications, making them an active area of research in materials science and electrochemistry.

In recent years, PEs have been drawing the attention of researchers due to their unique ability to provide high ionic conductivity and good mechanical stability in comparison to conventional liquid electrolytes. PEs have emerged as a promising class of materials for energy storage and conversion applications due to their potential to provide a wide range of properties that are critical for these applications. Among these properties, their optical and electrical characteristics are of particular interest, given their suitability for use in various energy storage and conversion devices. As a result, researchers have conducted numerous studies on various aspects of PEs, including their electrical conductivity, dielectric behavior, and mechanical properties, to better understand their potential for use in energy storage

applications [6–9]. By understanding these properties, researchers can optimize the design and composition of PEs for various applications in energy storage and conversion devices.

Polymer electrolytes have attracted attention for their potential use in light-emitting devices due to their optical and mechanical properties. By adjusting their chemical composition and processing conditions, polymer electrolytes can exhibit various optical characteristics such as the absorption, transmission, reflection, refraction, energy gap, dielectric constants, dielectric loss, and extinction coefficient.

PEs' optical properties are crucial for developing new optoelectronic devices. PEs are attractive for these devices because they are flexible, low-cost, and can be easily made into thin films. Studying the optical properties of PEs can help researchers understand their light–matter behavior and develop innovative applications. The use of polymers in optical devices has gained significant attention in recent years due to their low cost, flexibility, and ease of processing. Improving their optical properties is essential for their use in various applications such as polarizers, color filters, optical communication devices, biomedical devices, and data storage devices [10]. This has led to increased research interest in the development of novel polymer-based materials for use in optoelectronic devices such as polymer lasers, organic photovoltaic devices, and light-emitting diodes [11]. The industry has invested heavily in supporting these scientific efforts to develop new materials with improved optical properties.

Dopants such as semiconductor particles, metals, and salts have been used to modify the optical and electrical properties of certain polymers [12–15]. These modifications depend on the type of dopant and its concentration [16]. The irradiation (gamma ray, electron beam, ion, and plasma) of polymers can also modify the physical and chemical properties of polymers [17–21] by bond scission, the release of hydrogen, free radical formation, cross-linking, and various oxidation reactions [22–31].

Nouh et al. [32] investigated the effect of gamma radiation (5–25 kGy) on the structure and optical properties of polycarbonate-polybutylene terephthalate/silver nanocomposite films. The study found that gamma irradiation dispersed Ag nanoparticles within the matrix, resulting in an increase in the amorphous phase and a reduction in the optical energy gap. The nanocomposite also exhibited a color change response characterized by an increase in color intensity. This study suggested that gamma radiation may be a promising method for modifying the optical properties and structure of these nanocomposite films.

Aldaghri et al. [33] studied the impact of gamma irradiation on the chemical and optical properties of the conjugated copolymer B-co-MP. Various concentrations of B-co-MP were evaluated under gamma irradiation doses ranging from 5 to 20 kGy. The study found that gamma irradiation resulted in a significant blue shift in the absorption, fluorescence, and amplified spontaneous emission (ASE) spectra of the copolymer, indicating a widening in the energy gap and a reduction in the number of carbon atoms. These findings provide valuable insights into the effects of gamma irradiation on conjugated copolymers and their potential applications in optoelectronics. In their research, Forster et al. [34] explored the impact of gamma irradiation on luminescent films created by incorporating the $[Eu(tta)_3(H_2O)_2]$ complex into a polycarbonate matrix. The luminescent films were subjected to varying doses of gamma radiation, and alterations in their luminescence properties were observed. The research showed that the emission spectra of the films exhibited the characteristic band transitions of the $Eu^{3+}$ ion, suggesting the possibility of producing luminescent films. Additionally, the findings revealed that the doped system maintained its photoluminescence behavior even after gamma irradiation, indicating that the use of gamma irradiation did not impair the performance of the doped system.

Al-Kadhemy M. F. H. et al. [35] studied the effect of gamma radiation on the optical properties of PMMA/PS polymer blends. The researchers found that gamma rays caused a decrease in the energy bandgap values of the polymer blends, which could be attributed to the rearrangement of polymer molecules. Gamma irradiation was also found to influence the optical constants of the blends. These findings provide valuable insights into the potential use of gamma radiation to modify the optical properties of polymer blends.

This research aimed to enhance the optical properties of poly(ethylene oxide) (PEO) through two different methods: doping with different concentrations of potassium iodide (KI) and irradiation with gamma radiation. By incorporating KI as a dopant and the use of gamma radiation, it was expected to modify the electronic and optical properties of PEO, such as the energy gap, absorption, and other optical constants. The primary goal of this study was to explore the impact of these two approaches on PEO's optical properties to assess their suitability for use in various applications, including optoelectronics and optical devices. By gaining a deeper understanding of the impact of these modifications on PEO's optical properties, this study may pave the way for the development of new and improved materials for various industrial applications.

## 2. Materials and Methods

In this study, the researchers aimed to improve the optical properties of a poly(ethylene oxide) (PEO) resin by doping it with different concentrations of potassium iodide (KI) and irradiating it with gamma radiation. KI salts are often used in the preparation of polyethylene oxide (PEO) films as a dopant to improve the film's conductivity. KI salts are known to dissociate in water, resulting in the formation of $K^+$ and $I^-$ ions. When KI salts are added to a PEO solution, the $K^+$ ions act as a counterion, neutralizing the negative charges on the PEO chains, which can enhance the polymer's conductivity. Additionally, the presence of $I^-$ ions can facilitate the formation of complex ions with PEO chains, leading to further improvement in conductivity.

To prepare the electrolyte films, the PEO resin was first dissolved in acetone and then mixed with KI salt as a dopant. Different concentrations of KI (0, 2, 6, 10, and 15 wt%) were used in the preparation of the films. The PEO resin had an average molecular weight of 300,000 g/mol.

To ensure complete dissolution of the polymer, the mixture was stirred with a magnetic stirrer for 24 h. The prepared solution was then cast onto flat glass plates and left to dry in the ambient air at room temperature for 48 h. To remove any remaining solvent, the films were oven-dried at 40 °C for three days. The resulting solid films were on average 70 μm thick.

After preparation, the composite films were irradiated using a Co-60 gamma source irradiator from the Jordan Atomic Energy Commission. All samples were irradiated in air at room temperature by a Co-60 gamma source with a dose rate of 205.965 Gy/h The PEO samples were exposed to an absorbed dose of 100 Gy. One week after the first irradiation, they were exposed to an absorbed dose of 200 Gy.

To study the optical absorption spectra of each film, a Cary UV–Vis spectrophotometer was used at room temperature both before and after gamma irradiation. The resulting spectra were analyzed from 300 to 800 nm.

## 3. Results and Discussion

The following optical properties of the prepared PEO electrolyte films were investigated and analyzed.

### 3.1. Absorbance and Absorption Coefficient

The transmitted intensity *I* can be calculated using

$$I = I_0 \, e^{-\alpha(\omega) \, x} \tag{1}$$

where *x* is the film thickness and $I_0$ is the incident intensity, and the absorption coefficient $A(\omega)$ can be calculated in terms of the absorbance ($\alpha(\omega) = \log_{10}(I_0/I)$) [36–38]:

$$\alpha(\omega) = \frac{2.303}{x} \log_{10}(I_0/I) = \frac{2.303}{x} A(\omega) \tag{2}$$

In Figure 1a, it is observed that the optical absorption of the PEO electrolyte films increased significantly as the concentration of KI increased. The absorption decreased

rapidly with an increase in wavelength from 300 to 500 nm, followed by a slower decrease from 500 to 800 nm. The absorption spectra of all the doped samples had a peak between 330 and 480 nm, indicating the presence of a surface plasmon resonance peak. This peak was caused by the collective oscillation of valence electrons that were excited by resonant photons. The resonance occurred when the energy of the incident photons was equivalent to the natural frequency of the surface electron oscillations in opposition to the restoring force of the nuclei [39]. An increase in the KI concentration resulted in a redshift of the absorption maximum, which corresponded to a decrease in the energy gap [40].

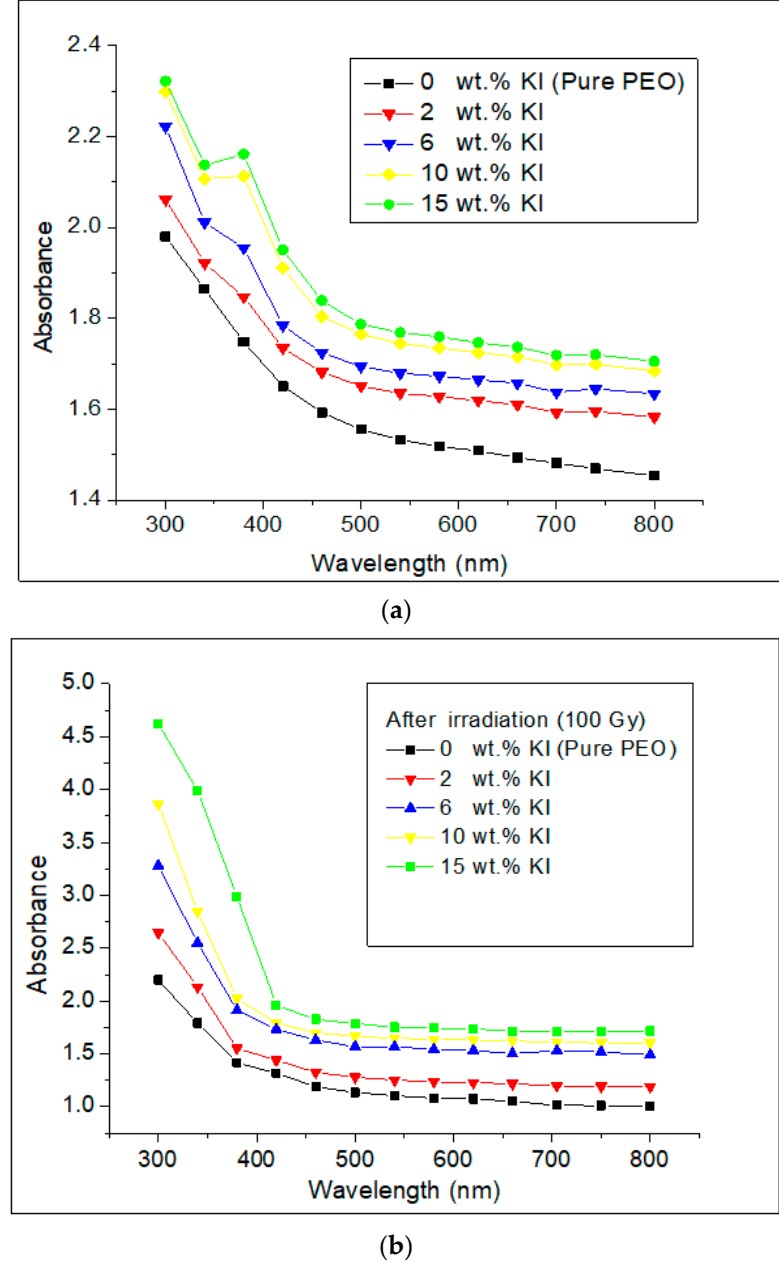

(a)

(b)

**Figure 1.** *Cont.*

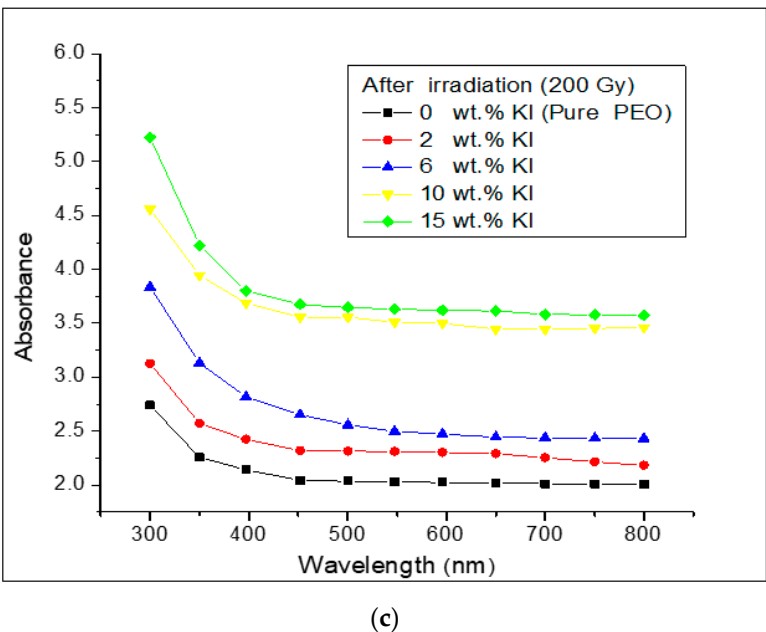

(**c**)

**Figure 1.** (**a**) The optical absorption spectra of PEO/KI thin films before gamma irradiation. (**b**) The optical absorption spectra of PEO/KI thin films after 100 Gy gamma irradiation. (**c**) The optical absorption spectra of PEO/KI thin films after 200 Gy gamma irradiation.

Figure 1b,c show that, after gamma irradiation, the absorption of the PEO electrolyte films at every KI concentration was higher than before irradiation, as shown in Figure 1a. This rise in optical absorption can be attributed to the impact of gamma radiation on the polymer composites, causing chain scission, which generated electronic levels in the energy gap [41]. The resulting electronic levels enhanced the absorption properties of the composites, leading to a considerable rise in optical absorption after gamma irradiation.

### 3.2. Energy Gap

The energy gap, which refers to the energy difference between the valence band and the conduction band in a material, can be characterized by the fundamental absorption that arises from the excitation of electrons from the valence band to the conduction band [42]. The size and type of the energy gap can provide useful information about the electronic structure of a material and its optical properties. For example, a narrow energy gap typically indicates that a material is more conductive, whereas a wide energy gap is typically associated with insulating behavior. The type of energy gap, whether direct or indirect, also affects the optical properties of a material, such as its ability to absorb and emit light. Therefore, investigating the energy gap in PEO electrolyte films is important for understanding their optical properties and potential applications.

The energy gap $E_g$ can be obtained from the Tauc formula [43]:

$$\alpha(\omega)\,\hbar\omega = B(\hbar\omega - E_g)^m \tag{3}$$

where $B$ is a constant related to band tailing, $\hbar\omega$ is the incident photon energy, and the exponent $m = 1/2$ relates to the allowed direct transition.

The energy gap of the prepared PEO electrolyte films was determined by analyzing their linear absorption spectra. To determine the energy gap, the plots of $(\alpha\hbar\omega)^2$ versus the incident photon energy $\hbar\omega$ were generated for the thin films, as shown in Figure 2a–c. The linear portions of the curves were then analyzed to determine the intercept with the $\hbar\omega$ axis, which provides the value of $E_g$.

In Figure 2a, the results revealed that the energy gap ($E_g$) decreased as the concentration of the KI dopant increased (as observed in Table 1). The reason behind this trend can be

attributed to the increase in disorder within the polymer structure [44]. The introduction of dopants can create charge transfer complexes in the polymer, thus forming ionic pathways, which leads to a narrowing of the energy gap [45].

**Table 1.** The optical energy band gap $E_g$ for PEO/KI thin films before and after gamma irradiation.

| PEO Thin Film | $E_g$ (eV) before Gamma Irradiation | $E_g$ (eV) after Gamma Irradiation | |
|---|---|---|---|
| | | 100 Gy | 200 Gy |
| 0 wt.% KI | 2.307 | 2.294 | 2.195 |
| 2 wt.% KI | 2.271 | 2.254 | 2.140 |
| 6 wt.% KI | 2.222 | 2.156 | 2.043 |
| 10 wt.% KI | 2.183 | 2.000 | 1.922 |
| 15 wt.% KI | 2.156 | 1.925 | 1.851 |

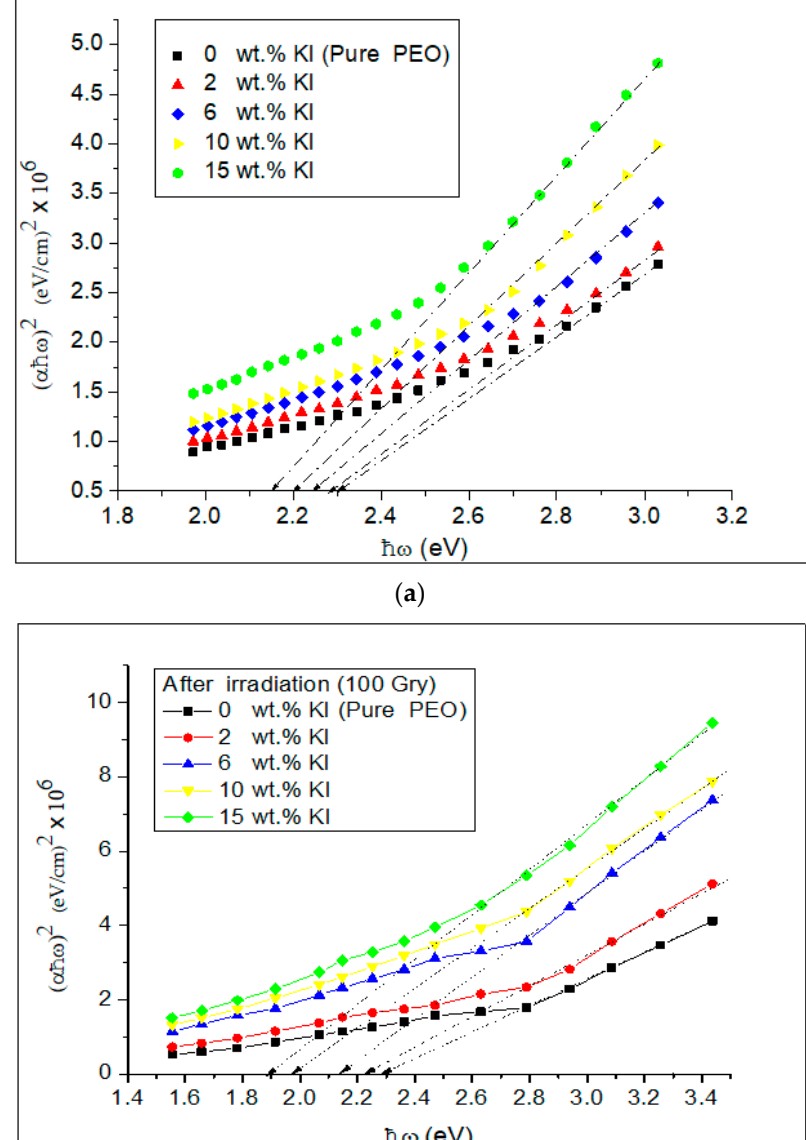

(a)

(b)

**Figure 2.** *Cont.*

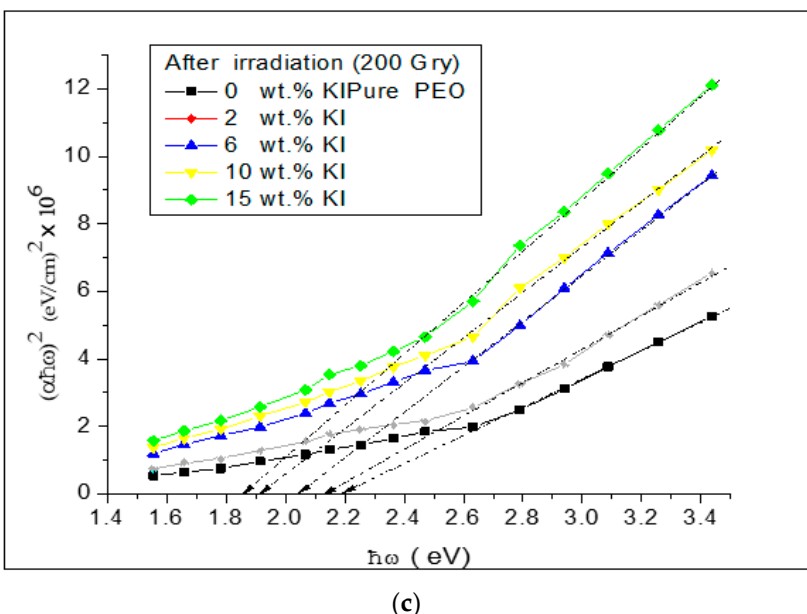

**(c)**

**Figure 2.** (**a**) The variation of $(\alpha\hbar\omega)^2$ with the incident photon energy ($\hbar\omega$) for PEO/KI thin films before gamma irradiation. (**b**) The variation of $(\alpha\hbar\omega)^2$ with the incident photon energy ($\hbar\omega$) for PEO/KI thin films after 100 Gy gamma irradiation. (**c**) The variation of $(\alpha\hbar\omega)^2$ with the incident photon energy ($\hbar\omega$) for PEO/KI thin films after 200 Gy gamma irradiation.

The energy gap values of the irradiated samples are presented in Table 1 and Figure 2b,c. It was observed that the energy gap values of the irradiated samples decreased significantly compared to the unirradiated samples (Figure 2a). This decrease in the energy gap value after irradiation can be attributed to the creation of new electronic levels in the energy gap or the formation of carbon-enriched clusters due to the release of hydrogen [41,46]. Table 1 provides a comparison of the energy gap values before and after irradiation. Overall, the irradiation process had a significant effect on the energy gap values of the PEO electrolyte films, resulting in a considerable decrease in the energy gap values.

*3.3. Refractive Index and Extinction Coefficient*

The refractive index (*n*) is a fundamental optical property of a material and is defined as the ratio of the speed (*c*) of light in a vacuum to the phase velocity (*v*) of light in a given medium.

$$n = c/v \tag{4}$$

In other words, the refractive index indicates how much a material slows down the speed of light as it passes through it. The refractive index can be calculated using Snell's law, which relates the angle of incidence and the refraction of a light ray as it passes through two media with different refractive indices. The refractive index of a material depends on various factors such as its chemical composition, density, temperature, and the wavelength of the light passing through it.

When light passes through a material, it is usually absorbed to some extent, which affects its refractive properties. To account for this absorption, a complex refractive index can be defined. The complex refractive index consists of two parts: the real part n, which determines the speed of light in the medium, and the imaginary part k, which describes the absorption of the material. The imaginary part k is related to the amount of light that is absorbed per unit length, and its value is typically small for transparent materials, but can be significant for highly absorbing materials such as metals.

$$n^* = n + i\,k \tag{5}$$

where $k$ is the extinction coefficient and relates to the absorption of light. It is defined as

$$k = \alpha \, \lambda / 4 \, \pi \tag{6}$$

where $\alpha$ is the absorption coefficient and $\lambda$ is the wavelength of the incident photons.

The refractive index $n$ is related to the extinction coefficient $k$ [47]:

$$n = \frac{1+R}{1-R} + \sqrt{\frac{4R}{(1-R)^2} - k^2} \tag{7}$$

where $R$ is the reflectance.

The extinction coefficient was found to increase with increasing KI concentration, as shown in Figure 3a. This indicated that the films became more absorbing as the concentration of KI increased.

Furthermore, the extinction coefficient was found to increase significantly with increasing gamma irradiation dose, as shown in Figure 3b,c. This increase in absorption can be attributed to the gamma-induced chain scission in the polymer composites, which creates electronic levels in the energy gap, as discussed earlier in Section 3.2.

The absorption coefficient $\alpha$, which is related to the imaginary part of the refractive index, is directly proportional to the extinction coefficient. Therefore, the increase in the extinction coefficient with increasing KI concentration and gamma irradiation dose is indicative of an increase in the absorption coefficient $\alpha$ and, thus, an increase in the amount of light absorbed by the PEO electrolyte films.

Figure 4a shows a significant increase in the refractive index $n$ with increasing KI concentration. In general, the refractive index of a material is related to its density, chemical composition, and the way in which its molecules interact with electromagnetic radiation. In this study, the increase in the refractive index with increasing KI concentration suggested that the addition of the dopant had a significant impact on the optical properties of the PEO electrolyte films. As discussed earlier, KI is known to create charge transfer complexes in the polymer and create ionic pathways, which may increase the density of the states in the energy gap and affect its refractive index.

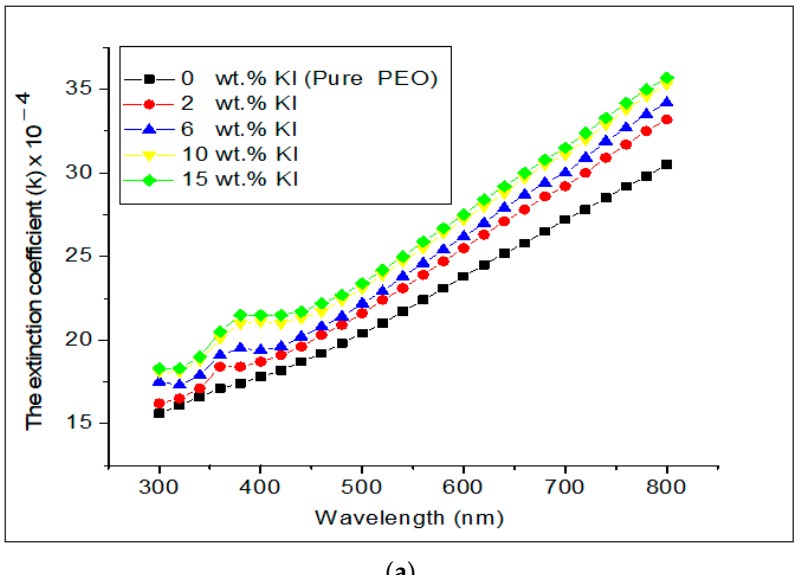

(**a**)

**Figure 3.** *Cont.*

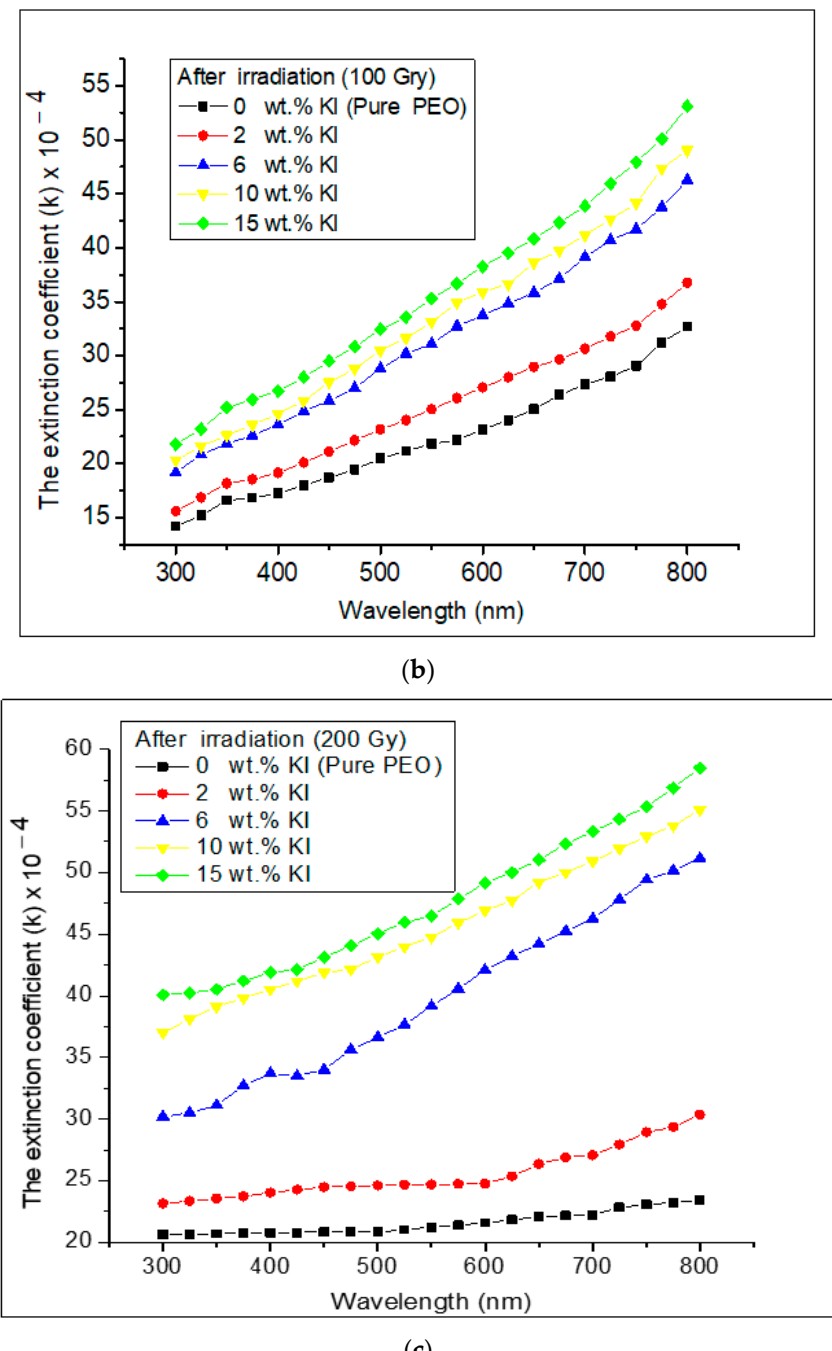

**Figure 3.** (**a**) The extinction coefficient (k) versus the incident photon wavelength for PEO/KI thin films before gamma irradiation. (**b**) The extinction coefficient (k) versus the incident photon wavelength for PEO/KI thin films after 100 Gy gamma irradiation. (**c**) The extinction coefficient (k) versus the incident photon wavelength for PEO/KI thin films after 200 Gy gamma irradiation.

Additionally, the increase in refractive index with irradiation dose (as shown in Figure 4b,c) indicates that gamma irradiation can alter the molecular structure of the PEO electrolyte films, which can influence their refractive properties. This effect can be attributed to the formation of new electronic states in the energy gap, which causes a decrease in the energy gap, leading to an increase in the refractive index.

It is also worth noting that there is an inverse relationship between the optical energy gap and the refractive index. This relationship can be explained by the fact that a smaller energy gap corresponds to a larger density of states, resulting in a higher refractive index.

Increasing the refractive index of a material is essential for many optical applications and devices [10]. For instance, optical sensors, which are widely used in medical and environmental monitoring, require high refractive index materials to enhance their sensitivity and accuracy. Similarly, mirrors and wave-guide optical circuits also require high refractive index materials to achieve efficient reflection and light confinement. In the field of photovoltaics, the refractive index of the anti-reflection coatings plays a crucial role in reducing the reflection losses and increasing the light absorption in the solar cells. Optical interference filters, which are used in many applications such as spectroscopy, require materials with a high refractive index to achieve high spectral selectivity. Therefore, increasing the refractive index of materials is of great significance in many fields of optics and photonics.

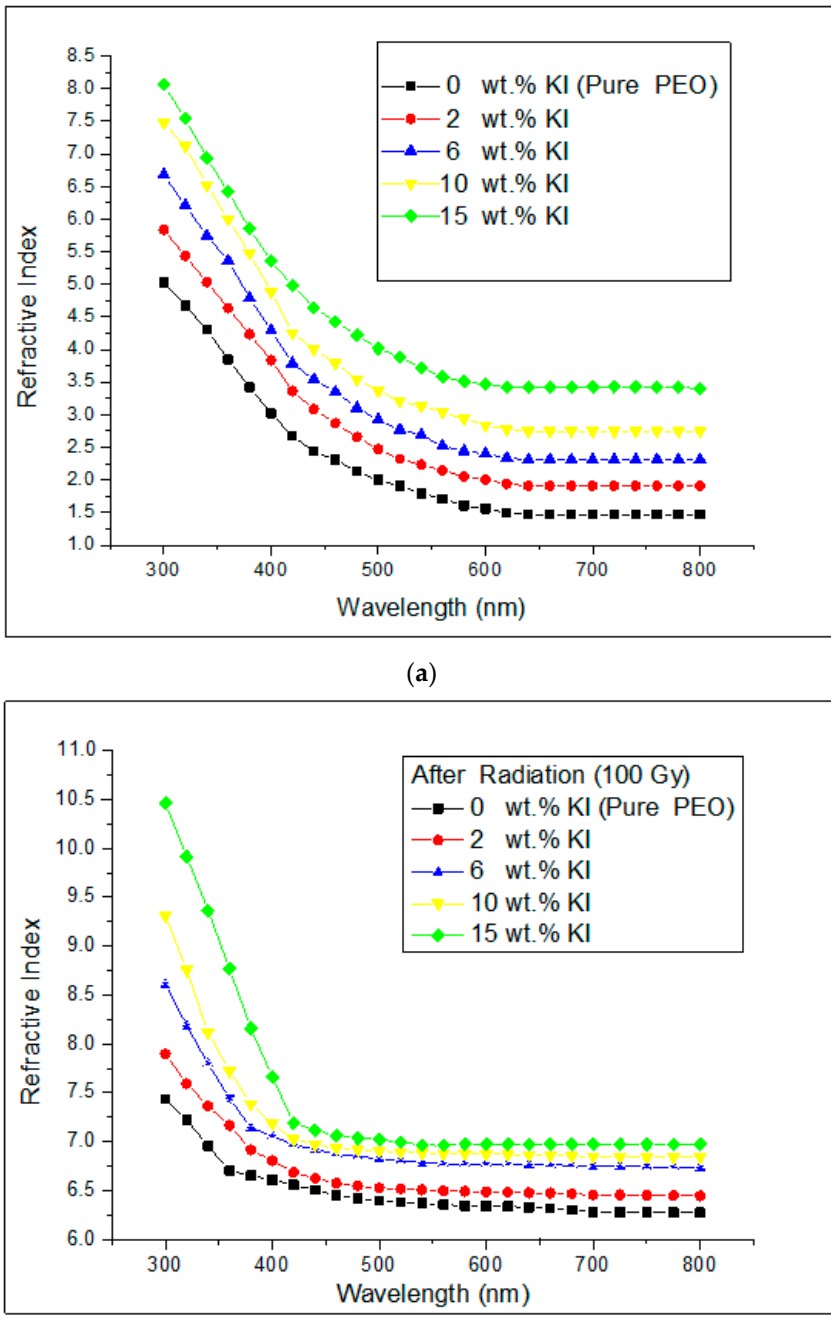

(a)

(b)

**Figure 4.** *Cont.*

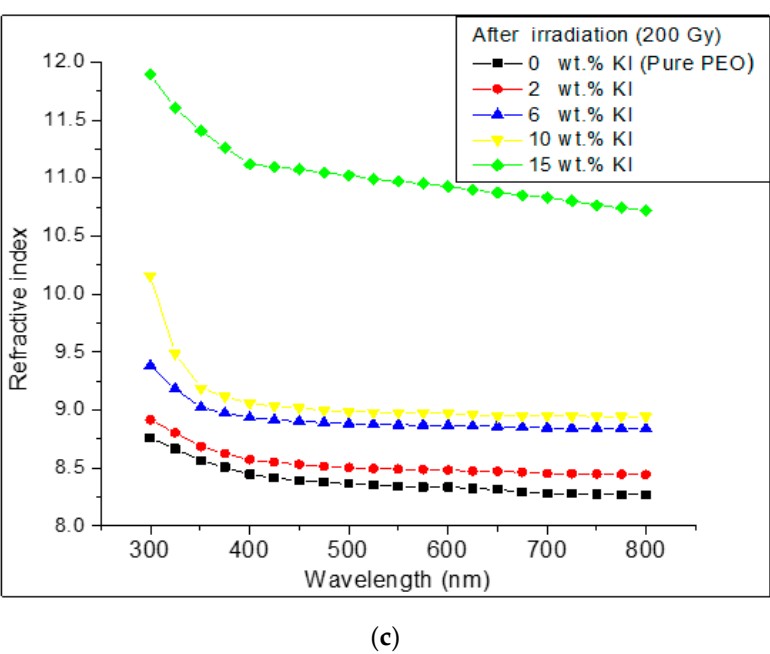

(**c**)

**Figure 4.** (**a**) The refractive index (n) versus the incident photon wavelength for PEO/KI thin films before gamma irradiation. (**b**) The refractive index (n) versus the incident photon wavelength for PEO/KI thin films after 100 Gy gamma irradiation. (**c**) The refractive index (n) versus the incident photon wavelength for PEO/KI thin films after 200 Gy gamma irradiation.

### 3.4. Dielectric Constant

The dielectric constant is a fundamental property of materials that characterizes their ability to polarize when exposed to an electric field. The dielectric constant, denoted by $\varepsilon$, can be expressed as the sum of its real part ($\varepsilon'$) and imaginary part ($\varepsilon''$).

$$\varepsilon = \varepsilon' + i\,\varepsilon'' \tag{8}$$

The real part of the dielectric constant describes the speed at which light propagates through a material, while the imaginary part, also known as the dielectric loss, quantifies the extent to which a dielectric can absorb energy from an applied electric field as a result of dipole motion. In other words, the dielectric loss determines the extent to which energy is lost as heat in a dielectric material when an electric field is applied [48].

The real part ($\varepsilon'$) and imaginary part ($\varepsilon''$) can be calculated in terms of the refractive index $n$ and the extinction coefficient $k$ from the following equations:

$$\varepsilon' = n^2 - k^2,\ \varepsilon'' = 2nk \tag{9}$$

Figure 5a displays a graph that illustrates the relationship between the real part of the dielectric constant ($\varepsilon'$) and the wavelength of the incident photon before subjecting the PEO film to gamma irradiation. It is evident from the graph that an increase in the KI concentration led to a rise in the real part of the dielectric constant ($\varepsilon'$). Subsequently, Figure 5b,c demonstrate that the real part of the dielectric constant ($\varepsilon'$) experienced a substantial increase as the irradiation dose increased.

Figure 6a shows a plot of the dielectric loss ($\varepsilon''$) as a function of the incident photon wavelength before gamma irradiation. As the KI concentration increased, the dielectric loss increased considerably, and a surface plasmon resonance (SPR) peak appeared in the absorption spectra of all doped samples in the range of approximately 330 to 480 nm. Additionally, the dielectric loss increased significantly with an increase in the gamma irradiation dose, as can be observed in Figure 6b,c.

The process of gamma irradiation led to the breaking of chains in polymer composites, resulting in various changes such as cross-linking, the release of hydrogen, and the absorption of oxygen by carbon [41]. This caused the polymer composites to become more polar, leading to the formation of new states in the energy gap, known as interband states [49]. The formation of these interband states reduced the energy gap and increased the dielectric constant.

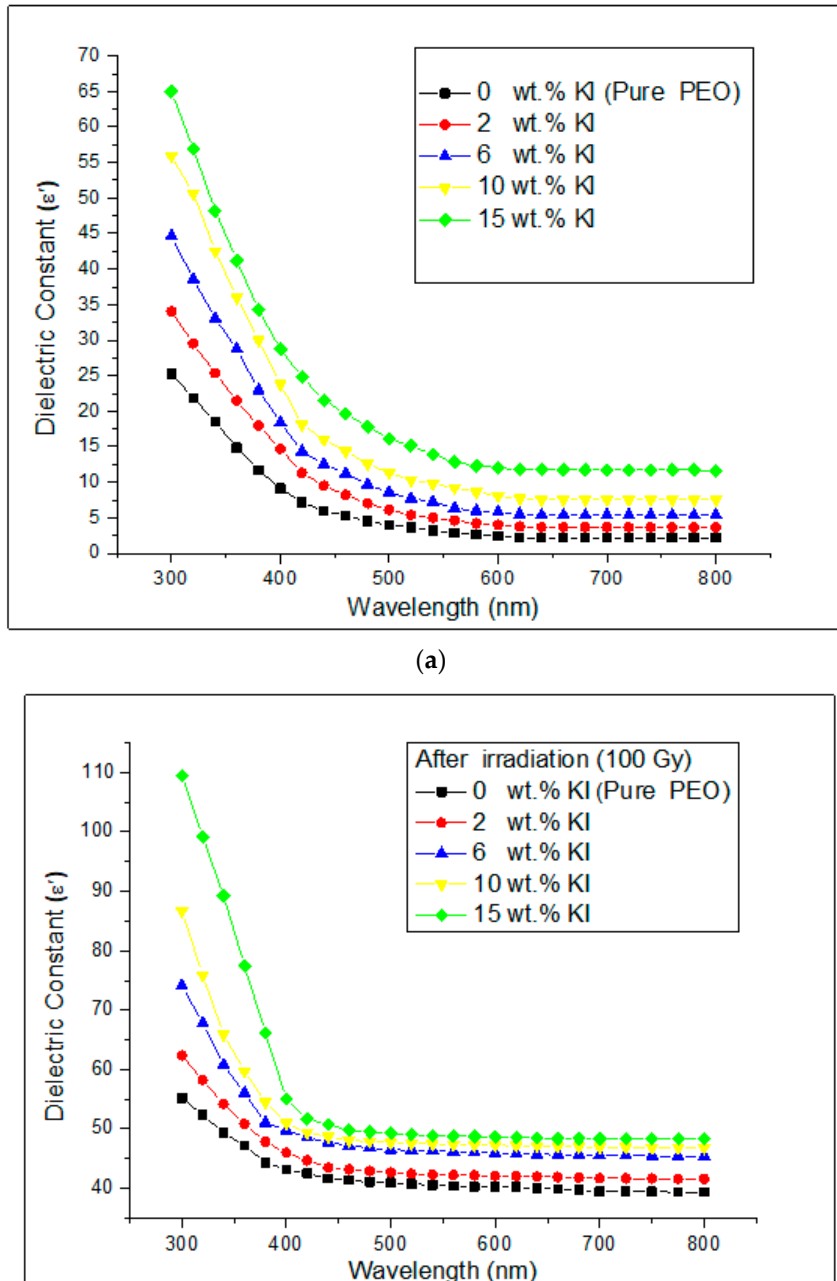

(a)

(b)

**Figure 5.** *Cont.*

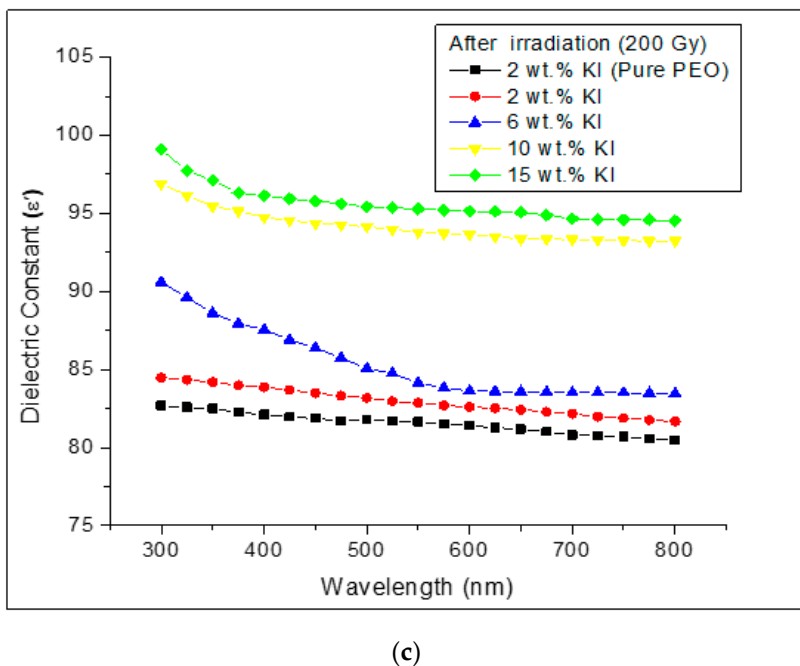

(**c**)

**Figure 5.** (**a**) The real part of the dielectric constant versus the incident photon wavelength for PEO/KI thin films before gamma irradiation. (**b**) The real part of the dielectric constant versus the incident photon wavelength for PEO/KI thin films after 100 Gy gamma irradiation. (**c**) The real part of the dielectric constant versus the incident photon wavelength for PEO/KI thin films after 200 Gy gamma irradiation.

The Penn model [50] is a theoretical model proposed to describe the relationship between the dielectric constant and the energy gap of a material. According to this model, there is an inverse correlation between the two parameters, meaning that an increase in the energy gap leads to a decrease in the dielectric constant, and vice versa. The model was proposed by Penn in 1962 and has been widely used in the field of solid-state physics to explain the optical and electronic properties of materials.

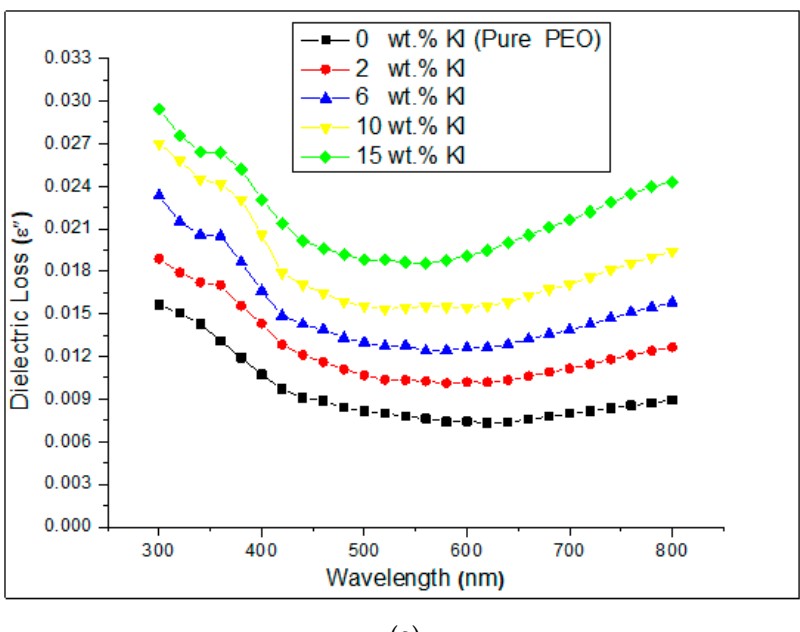

(**a**)

**Figure 6.** *Cont.*

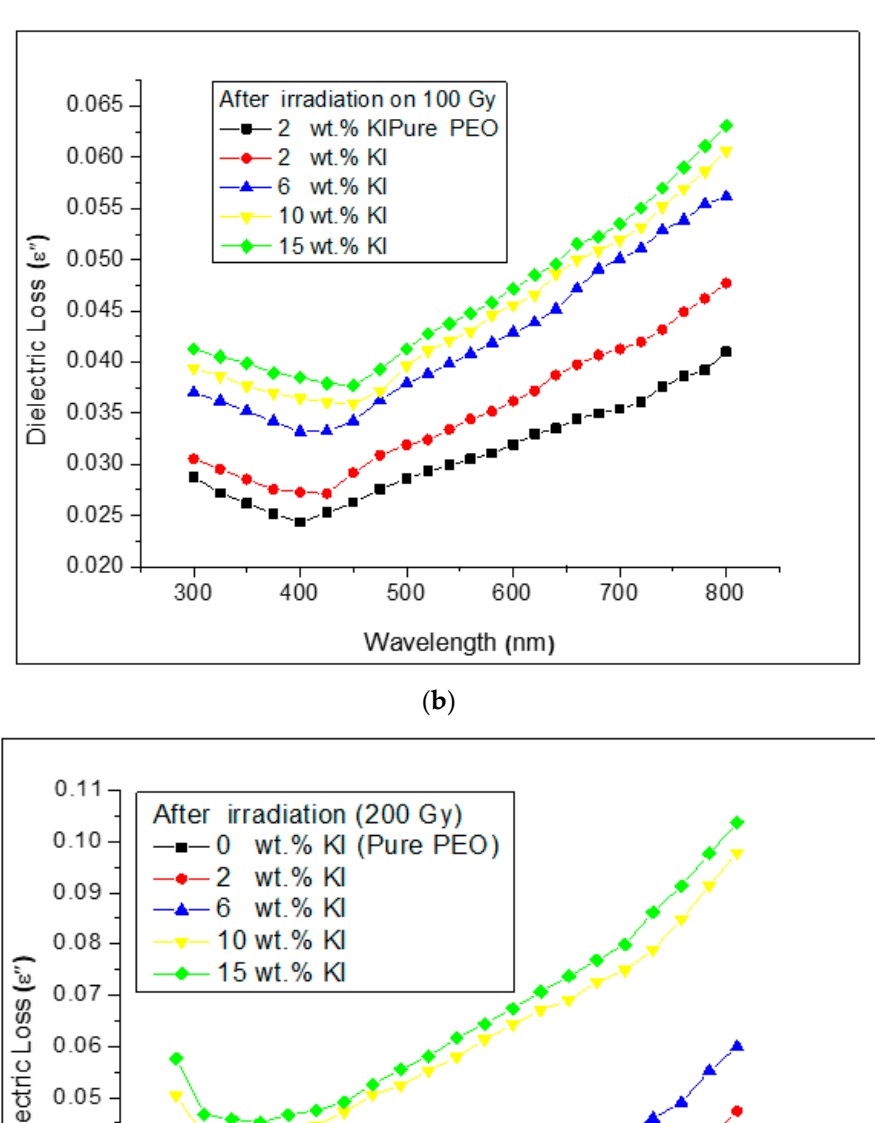

(**b**)

(**c**)

**Figure 6.** (**a**) The imaginary part of the dielectric constant versus the incident photon wavelength for PEO/KI thin films before gamma irradiation. (**b**) The imaginary part of the dielectric constant versus the incident photon wavelength for PEO/KI thin films after 100 Gy gamma irradiation. (**c**) The imaginary part of the dielectric constant versus the incident photon wavelength for PEO/KI thin films after 200 Gy gamma irradiation.

Aziz et al. [51] conducted an experimental study to verify the validity of the Penn model. Their results showed that the Penn model accurately described the relationship between the energy gap and the dielectric constant of these materials.

For energy-storage devices, increasing the dielectric constant of the electrolyte is crucial since it leads to an increase in their energy capacitance and breakdown field. However, it is also important to maintain low dielectric loss in these applications. Achieving both a high

dielectric constant and low dielectric loss can be challenging and requires a compromise between the two factors [39]. This is because materials with high dielectric constants often have high dielectric losses as well, and vice versa. Therefore, researchers need to strike a balance between these properties to optimize the performance of energy-storage devices. Achieving this balance can be accomplished through the careful selection of materials and their processing conditions. For instance, the use of nanocomposites, which consist of a polymer matrix filled with nanoparticles, can enhance both the dielectric constant and the dielectric loss, enabling a good balance between the two properties. Furthermore, researchers can modify the chemical structure of the polymer or use a mixture of different polymers to obtain the desired properties.

## 4. Conclusions

In summary, the research conducted in this study demonstrated that the optical properties of thin PEO electrolyte films can be altered through the process of doping with KI salt and gamma irradiation. By increasing the concentration of KI and the dose of gamma irradiation, it is possible to lower the energy gap and increase the absorption, refractive index, and dielectric constant of the PEO films. These modifications hold great promise for optimizing the use of polymers in various optical applications and energy-storage devices. The findings of this study suggested that, by manipulating the optical properties of thin PEO films, it is possible to achieve the desired characteristics for specific applications, while still maintaining low dielectric loss. The results of this research contribute to a deeper understanding of the relationship between the structure and properties of PEO electrolyte films and offer insights into the development of new materials with improved optical characteristics.

**Author Contributions:** Conceptualization, A.A.H.Q., B.A.A.S. and M.A.-T.; methodology, A.A.H.Q. and M.A.-T.; software, A.A.H.Q.; validation, Z.M.E., H.K.J.; formal analysis, A.A.H.Q. and M.A.-T.; investigation, R.I.A.; resources, M.A.-T.; data curation, A.A.H.Q. and M.A.T.; writing—original draft preparation, A.A.H.Q. and M.A.-T.; writing—review and editing, A.A.H.Q. and M.A.T.; visualization, A.A.H.Q.; supervision, A.A.H.Q.; project administration, A.A.H.Q.; funding, A.A.H.Q. All authors have read and agreed to the published version of the manuscript.

**Funding:** This research received no external funding.

**Data Availability Statement:** The data that support the findings of this study are available on request from the corresponding author.

**Acknowledgments:** The authors thank the Jordan Atomic Energy Commission for the use of their facilities to irradiate the samples.

**Conflicts of Interest:** The authors declare no conflict of interest.

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
