# Peer review of "Effects of Gamma Irradiation on Optical Properties of Poly(ethylene oxide) Thin Films Doped with Potassium Iodide"

_jcs, doi:10.3390/jcs7050194_

Round 1
Reviewer 1 Report
Authors describe the modification of a PEO film through gamma irradiation. PEO modification are studied using UV-VIS absorption spectroscopy. I suggest to publish after minor revisions.
Comments:
1) Figure 1.c is upset please correct.
2) Please increase the quality of plots, Some plots like 2c result unfocused.
3) Please extend materials and method parts, describing how they conducted gamma irradiation
4) The introduction part is too long, please reduce references descriptions.
5) Describe the role of KI salts in the preparation of PEO films
Author Response
Dear Reviewer,
Thank you very much for your thorough evaluation of our manuscript and for providing insightful comments. We appreciate the time and effort you put into reviewing our work. We have carefully considered all of your suggestions and have made the necessary changes to the manuscript accordingly.
We are pleased to inform you that your recommendations have been taken into consideration, and we believe that these revisions have significantly improved the quality of our work. We are grateful for your valuable feedback, which has helped us to strengthen our study.
Thank you again for your helpful comments and for recommending our manuscript for publication.
Best regards
Reviewer 2 Report
The paper of Qwasmeh et al. reports the effect of gamma irradiation on the optical properties of thin films of poly(ethylene oxide) (PEO) doped with different concentrations of potassium iodide. The article is written quite clearly and the experimental results are also quite interesting and have been consistently reported.
Some English and typing errors are present in the manuscript. They should be corrected. In particular Figure 1c appears to be upside down.
Therefore, I can recommend the paper for publication on J. Compos. Sci. after minor revision.
Author Response
Dear Reviewer
Thank you very much for your thorough evaluation of our manuscript and for providing insightful comments. We appreciate the time and effort you put into reviewing our work. We have carefully considered all of your suggestions and have made the necessary changes to the manuscript accordingly.
Best regards
Round 2
Reviewer 1 Report
Authors discussed and answered to all my comments. I reccomend to publish in this form.
Reviewer 2 Report
In my opinion, the article can be accepted in present form.